# Human Perception of Birds in Two Brazilian Cities

Gabriela Rosa Graviola [1] , Milton Cezar Ribeiro [1,2] and João Carlos Pena [1,*]

1 Spatial Ecology and Conservation Lab (LEEC), Department of Ecology, Instituto de Biociências, São Paulo State University (UNESP), Rio Claro 13506900, SP, Brazil; gabriela.rosa@unesp.br (G.R.G.); milton.c.ribeiro@unesp.br (M.C.R.)
2 Centro de Estudos Ambientais (CEA), São Paulo State University (UNESP), Rio Claro 13506900, SP, Brazil
* Correspondence: joaocpena@gmail.com; Tel.: +55-11-98123-2180

**Simple Summary:** Wildlife has coexisted with human society for thousands of years. While there is a general consensus that nature in urban areas should be increased, the way people perceive the animals that live within cities varies greatly. Considering research on this issue, there are very few studies analyzing people's perception of urban wildlife, and the list of animals investigated is still very limited; the lowest frequency of these studies is in South America. In order to fill this gap, we aimed to identify how people perceive birds in two Brazilian cities. We observed that people can recognize the most frequent bird species and are aware of the ecological importance of birds. We also showed that most people associate most bird species with positive feelings such as beauty, joy, well-being, and peace. On the other hand, exotic species are considered pests and generate negative perceptions. Understanding how humans perceive animals plays a significant role in comprehending the contemporary human-nature relationship. Providing this knowledge is essential for planning environments where humans and animals interact and to garner broad support for biodiversity conservation in cities.

**Abstract:** Understanding how humans perceive animals is important for biodiversity conservation, however, only a few studies about this issue have been carried out in South America. We selected two Brazilian cities to assess people's perceptions of birds: Bauru (São Paulo, Brazil) and Belo Horizonte (Minas Gerais, Brazil). From the available bird data for each city, we developed a questionnaire and applied it between September 2020 and June 2021. The data obtained were analyzed by simple counts, a Likert scale, and percentages. Also, human feelings related to birds were placed on the Free Word Cloud Generator website. Our study confirmed that most respondents were aware of the importance of birds to ecological balance and that respondents had a generally positive attitude towards most of the bird species. However, they disliked exotic species such as the Domestic Dove and the House Sparrow, which are associated with disease, dirt, and disgust. Respondents also underestimated the number of birds that can live in urban areas and the song of birds is still a sense less experienced and perceived by people. Understanding these human–biodiversity relationships can help guide public policies and environmental education activities.

**Keywords:** urban wildlife; human–biodiversity relationships; questionnaires; sensations; urban ecology

## 1. Introduction

Wildlife has coexisted with urban environments for thousands of years [1]. Cities are unique ecosystems [2,3], where biodiversity is fundamental for the delivery of important ecosystem services such as water protection, heat island effect reduction, floods, and noises and air pollution [4,5]. Also, nature has positive effects on human well-being and health [6–9]. Since most humans live in urban regions, cities are the prime places where people can experience nature daily [10]. However, while there is a consensus that nature in urban areas should be increased, the way people perceive the animals that live within cities varies greatly. Human

perceptions of wildlife encompass a wide spectrum of emotions, ranging from admiration and respect to fear or even hatred [11]. Different avian characteristics elicit various responses in people, influenced by the identity and taxonomic kingdom of each species [12].

Animals evoke a range of responses from people: in general, birds, mammals, and amphibians/reptiles are liked most, while the attitude towards arthropods and other invertebrates is less positive among people [10]. However, there are some exceptions: mammals such as coyotes may be perceived with either indifference or fear [13] and rats are the least appreciated mammals by people [14]. On the other hand, insects such as butterflies are also popular, different from others such as cockroaches [15]. Furthermore, increasing familiarity with animals not only increases the range of attitudes towards them, but those attitudes may become more intense, either positive or negative [10].

Perception is linked to sensations, while attitude is a cultural posture formed by a long succession of perceptions [16]. Five senses allow humans to perceive and experience the world: sight, smell, taste, hearing, and touch. The way we interpret and apprehend the information transmitted by our senses and sensations in the world we live in is called perception [17]. Perception is the way an individual observes, understands, and interprets a referent object, action, or experience [18]. Our perception is based on our experience and is also on a myriad of other factors related to collective attributes (e.g., gender, race), values, norms, beliefs, preferences, and knowledge [19]. Therefore, the study of local people's perceptions and attitudes toward urban wildlife is important for guiding public policies related to environmental education, for example. By understanding how different groups of animals are perceived by society, it is possible to develop targeted educational strategies to address knowledge gaps about the importance of each species in maintaining the balance of ecosystems.

Considering research on this issue, there are very few studies analyzing people's perceptions of urban wildlife, and the list of animals investigated is still very limited [10,14,15]. A systematic review identified several knowledge gaps: more than 80% of the studies about how urban wildlife is perceived by people have involved mammals (only three studies) and the lowest frequency of these studies is in South America [11]. Furthermore, other research about positive interactions between humans and nature primarily took place in North America, Europe, Australasia, China, and Japan, with a limited number of studies also conducted in South America [20]. Among birds, this systematic review found only one study about people's perception of them [11]. A recent study conducted in Australia found that most residents had a generally positive attitude towards all birds [21]. Birds are of vital importance for ecological balance: they are responsible for seed dispersal and pollination, the control of insect populations, and assist in the balance of the food chain as predators and prey [22–25]. Even though birds are essential to ecological balance, how do people perceive them?

In this study, we aim to identify how people perceive birds in two Brazilian cities: Bauru (São Paulo State) and Belo Horizonte (Minas Gerais State). Understanding how humans perceive animals in cities plays a significant role in comprehending the contemporary human-nature relationship. Filling this knowledge gap is essential for planning urban landscapes where humans and animals can interact without conflicts and to garner broad support for biodiversity conservation in urban areas [2,10,11]. Several studies report that more intimate contact with nature can increase peoples' tolerance towards biodiversity and the willingness to protect it [15,26].

## 2. Materials and Methods

### 2.1. Study Areas and Bird Data

We assessed people's perceptions of birds in two Brazilian cities, Bauru (São Paulo State) and Belo Horizonte (Minas Gerais State). Both cities were chosen because of the availability of bird data collected by the authors mainly on the streets using similar survey methods. Furthermore, both cities are located in the transition zone between the Atlantic Forest and the Brazilian Savanna (Cerrado). Thus, it would be possible to assess how people perceive bird species they may encounter daily based on data on the bird communities

that occupy each city. Furthermore, both cities present specificities that may influence how people perceive urban bird communities.

Bauru is a medium-sized city in the Central-West region of the state of São Paulo (22°18′ S, 49°30′ W), with about 379,146 inhabitants and 567.85 hab/km$^2$ [27]. It is one of the cities in the western part of the state of São Paulo with a prosperous and well-developed economy. The city harbors 296 bird species and its urban landscape presents high heterogeneity, with streams, rivers, forest fragments, and important urban parks [28].

Belo Horizonte (19°55′ S, 43°56′ W) is the capital of the state of Minas Gerais and was one of the first planned cities in Brazil, which means that it is a city that was consciously designed through a systematic urban planning process before being built. In contrast to cities that develop organically over time, Belo Horizonte was intentionally laid out with specific considerations for infrastructure, land use, transportation, and overall urban design. It is the fourth richest Brazilian city, contributing 1.46% to the national GDP, trailing only behind São Paulo, Rio de Janeiro, and Brasília, respectively. The city harbors 370 bird species [27,29]. According to the last census, the population stands at 2,315,560 inhabitants and its population density is 6988.18 hab/km$^2$ [27].

In Bauru, we conducted bird observations using 10-min point counts, in which individual points were selected and the observer stopped at predefined spots, recording all the birds seen or heard for a predetermined time [30–32]. The records were taken during the mornings with favorable weather between December 2018 and March 2019 (the summer season in Brazil) and from September to December 2019 (spring season). This period coincides with the breeding period of most species in southeastern Brazil, including migratory species [23]. We established 36 sampling points (24 on streets and 12 in parks) and each point was visited six times, three during the summer and three during the spring, totaling 216 field observations (108 by season).

For Belo Horizonte, we used bird data previously published in the literature that were also collected across the streets of the southern region of the city using a similar protocol, which was used as a reference for the sampling design applied in Bauru [30,33]. The authors carried out bird surveys in 60 sample points, each with 3 replications, totaling 180 field observations. The selection of the survey points aimed to represent the variation of the influences of the streets and arboreal and herbaceous vegetation across the study area. Each survey point was at least 200 meters away from each other [30]. The authors conducted the fieldwork during the first three hours of daylight on days with favorable weather (sunny and non-windy days), only on working days to avoid great variation in people and vehicles in circulation. The bird survey was between September 2014 and January 2015, the period that also coincides with birds' breeding season in southeastern Brazil [23,30]. In both cities, we considered as 'frequency' (f) the number of times each species was recorded during the survey. We did not consider bird abundance in this study, because some species have a gregarious behavior (such as *Columba livia*), while others are observed singly or in pairs (such as *Pitangus sulphuratus*).

### 2.2. Assessing Human Perception of Urban Birds

From the bird data that we gathered, we developed a questionnaire for each city to analyze how the human population perceives the local avifauna. We carried out an opinion survey between September 2020 and June 2021, with the use of an online questionnaire that we created using the Google Forms tool [10]. Questionnaires were prepared with open and choice-structured questions: open responses are those in which the respondent answers in their own words, while choice-structured questions are structured in the form of a choice of some answer alternatives [34]. Questionnaires are considered an appropriate method because they limit the range of answers a participant can give and allow for a standardization of results [10].

We sent the questionnaires by e-mail and through Facebook and WhatsApp groups of Bauru and Belo Horizonte residents. On Facebook, there are city-specific groups that serve as platforms for discussing and debating various topics related to the city. Some of these

groups have a member numbers ranging from 100,000 to 300,000 people. In the same way, on WhatsApp, some communities focus on city-related topics. We joined these groups to share the questionnaires with their members. We also shared the questionnaires by email using institutional addresses mainly from researchers of both cities.

The answers to the questionnaires were anonymous, but the first part of the questionnaire asked for sociodemographic information regarding gender, age, education, and family income. These data are important to understand the social context that was achieved. In the second part, we addressed the following open and choice structured questions:

Open questions:

1. Describe in one word how you feel about urban birds
2. What do you believe is a benefit of birds in cities?
3. Do you believe that there is harm caused by birds in cities?

Choice structured questions:

4. How many different birds do you think you have seen within the urban area of Bauru?
5. 371 bird species have already been recorded within the entire territory of Belo Horizonte (which includes forested, rural, and urban areas)/ 276 bird species have already been recorded within the entire territory of Bauru (which includes forested, rural and urban areas). How many do you believe are capable of living in the urban area?

We also used a Likert scale [10,35–37], one of the most popular methods for conducting opinion surveys, in which, from a self-descriptive statement, the respondent chooses as a response option a scale of points with verbal descriptions that include extremes—such as "strongly agree" and "strongly disagree" [37,38]. Statements were created regarding the contribution of birds in ecological processes such as seed dispersal, pollination, and pest control. Thus, the respondents should express their degree of agreement with each sentence. We used 5 points on a scale from "strongly agree" to "strongly disagree". We addressed the following statements:

1. Birds contribute to seed dispersal.
2. Birds contribute to plant pollination.
3. Birds contribute to the control of pests, insects, and other animals.
4. Birds contribute to the prevention of the incidence of diseases.

Finally, we presented images of 20 birds (the 15 most frequent and the 5 least frequent in each city) and songs of the 6 most frequent species at each city. For these questions, respondents needed to mark which species they had seen and/or which sounds they had heard across the city.

### 2.3. Analyzing Peoples' Perceptions towards Birds

First, we analyzed the sociodemographic information and created a table with the main characteristics of the respondents participating in our research (Table 1). Data obtained from closed and Likert scale questions were analyzed by simple counts and percentages [39]. We also analyzed if the species most commonly observed and heard by people were also the most frequently recorded species. In this context, 'frequency' (f) refers to the number of times each species was recorded during the survey, considering all sampled points for each city. Therefore, frequency indicates the number of times the species was sighted during the surveys.

In the case of the open questions, we carried out two different approaches: to analyze the answers to the question "describe in one word how you feel about urban birds", the words were placed in the Free Word Cloud Generator website (https://www.freewordcloudgenerator.com/, accessed on 15 December 2022), which is a tool that creates a "cloud" of words, highlighting those that appear most frequently. This website also points out how many times each word was mentioned in each city. To analyze open questions 2 and 3, we grouped answers according to the harm or benefit mentioned and carried out simple counts and percentages to find out the main kinds of harm or benefit that were outlined by respondents.

**Table 1.** Characteristics of the respondents participating in the research about human perception of birds in Bauru and Belo Horizonte (Brazil).

| Variable | | Cities | |
|---|---|---|---|
| | | Bauru | Belo Horizonte |
| Gender | | | |
| | Male | 41 (36.6%) | 55 (44.7%) |
| | Female | 71 (63.4%) | 66 (53.7%) |
| | Other | 0 (0%) | 2 (1.6%) |
| Age | | | |
| | Up to 25 years | 33 (29.5%) | 16 (13.0%) |
| | 26 to 35 years | 36 (32.1%) | 44 (35.8%) |
| | 36 to 45 years | 12 (10.7%) | 32 (26%) |
| | 46 to 60 years | 22 (19.6%) | 17 (13.8%) |
| | 61 to 74 years | 8 (7.1%) | 14 (11.4%) |
| | More than 75 years | 1 (0.9%) | 0 (0%) |
| Education | | | |
| | Elementary and middle school | 2 (1.8%) | 0 (0%) |
| | High school | 9 (8.0%) | 4 (3.3%) |
| | Bachelor study incomplete | 29 (25.9) | 12 (9.8%) |
| | Bachelor study complete | 24 (21.4%) | 36 (29.3%) |
| | Master and doctorate degree | 47 (42.0%) | 71 (57.6%) |
| | Post-doctoral degree | 1 (0.9%) | 0 (0%) |
| Family Income | | | |
| | BRL 1000 reais or less | 2 (1.8%) | 1 (0.8%) |
| | BRL 1001 to 3000 | 36 (32.1%) | 18 (14.6%) |
| | BRL 3001 to 5000 | 29 (25.9%) | 20 (16.3%) |
| | BRL 5001 to 10,000 | 25 (22.3%) | 31 (25.2%) |
| | More than BRL 10,000 | 16 (14.3%) | 47 (38.2%) |
| | No answser | 4 (3.6%) | 6 (4.9%) |

## 3. Results

In Bauru, 112 responses were obtained, with 63.4% self-reporting that they were female and 36.6% reporting that they were male. Regarding education, 97.3% of respondents had undergraduate degrees, of which 41.0% also had postgraduate degrees. Regarding income, 32.1% of respondents had a low income, 25.9% had a medium income, 22.3% had a upper-middle income, and 14.3% had a high income.

In Belo Horizonte, 123 responses were obtained, with 53.7% self-reporting that they were female, 44.7% reporting that they were male, and 1.6% declaring themselves as of non-binary gender. Regarding education, 93.6% of respondents had undergraduate degrees, of which 51.0% also had a postgraduate degree. Regarding income, 14.6% of respondents had a low income, 16.3% had a medium income, 25.2% had an upper-middle income, and 38.2% had a high income. Most interviewed people were between 26 and 35 years old in both cities (Bauru: 32.1% and Belo Horizonte: 35.8%) and the minority was more than 75 years old.

Despite the questionnaires being sent to various email addresses and posted in groups that encompass a significant portion of the population in these cities (considering different social groups and classes), the vast majority of individuals who volunteered to respond to our questionnaire were individuals with high level of education (Table 1).

In Bauru, 91.1% of respondents strongly agree that birds have a fundamental role in seed dispersal, 77.7% that birds contribute to plant pollination, and 83% that birds

contribute to the control of pests, insects, and other animals (Figure S1). In Belo Horizonte, these values were 94.3%, 86.2%, and 86.2%, respectively (Figure S1).

In the case of open questions, when people were asked if they believed there was damage caused by birds in cities, most said yes (Bauru: 66% and Belo Horizonte: 70%). When asked about the type of damages, most people mentioned disgust and worry about the diseases that Domestic Pigeons (*Columba livia*) can transmit. Other problems/discomfort mentioned by people was electrical wiring damages, noise, dirt, and worry concerning ecological imbalance—in a situation of an uncontrolled population of pigeons and the presence of other exotic birds (Figure S2).

When people were asked which benefits they attribute to birds in cities, most respondents mentioned seed dispersal (Bauru: n = 36; Belo Horizonte: n = 29); biological control (Bauru: n = 24; Belo Horizonte: n = 35); life, joy and well-being (Bauru: n = 24; Belo Horizonte: n = 26) and beauty (Bauru: n = 21; Belo Horizonte: n = 19). The category "other" represents all different benefits that were mentioned only a few times such as: hope (n = 1), environmental indicators (n = 2), environmental education (n = 2), spiritual connection (n = 3), and biodiversity (n = 5; Figure S3).

In Bauru, the birds that are most daily seen by people were the Eared Dove (*Zenaida auriculata*), Domestic Pigeon (*Columba livia*), Great Kiskadee (*Pitangus sulphuratus*), House Sparrow (*Passer domesticus*), and White-eyed Parakeet (*Psittacara leucophthalmus*). Four of those species are also the ones most frequently recorded during our surveys (*P. sulphuratus*, 149 records; *Z. auriculata*, 141; *P. leucophthalmus*, 139; *P. domesticus*, 136; Figure 1A).

## A) Bauru

f = 141
**98.2%**
*Zenaida auriculata*

f = 54
**93.8%**
*Columba livia*

f = 149
**92.9%**
*Pitangus sulphuratus*

f = 136
**88.4%**
*Passer domesticus*

f = 139
**84.8%**
*Psittacara leucophthalmus*

f = 123
**74.1%**
*Patagioenas picazuro*

f = 41
**71.4%**
*Vanellus chilensis*

f = 77
**66.1%**
*Tyrannus melancholicus*

f = 47
**58.9%**
*Furnarius rufus*

f = 89
**55.4%**
*Brotogeris chiriri*

f = 46
**42.9%**
*Turdus leucomelas*

f = 83
**35.7%**
*Tangara sayaca*

f = 132
**29.5%**
*Pygochelidon cyanoleuca*

f = 37
**27.7%**
*Troglodytes musculus*

f = 46
**17.9%**
*Euphonia chlorotica*

**Figure 1.** *Cont.*

## B) Belo Horizonte

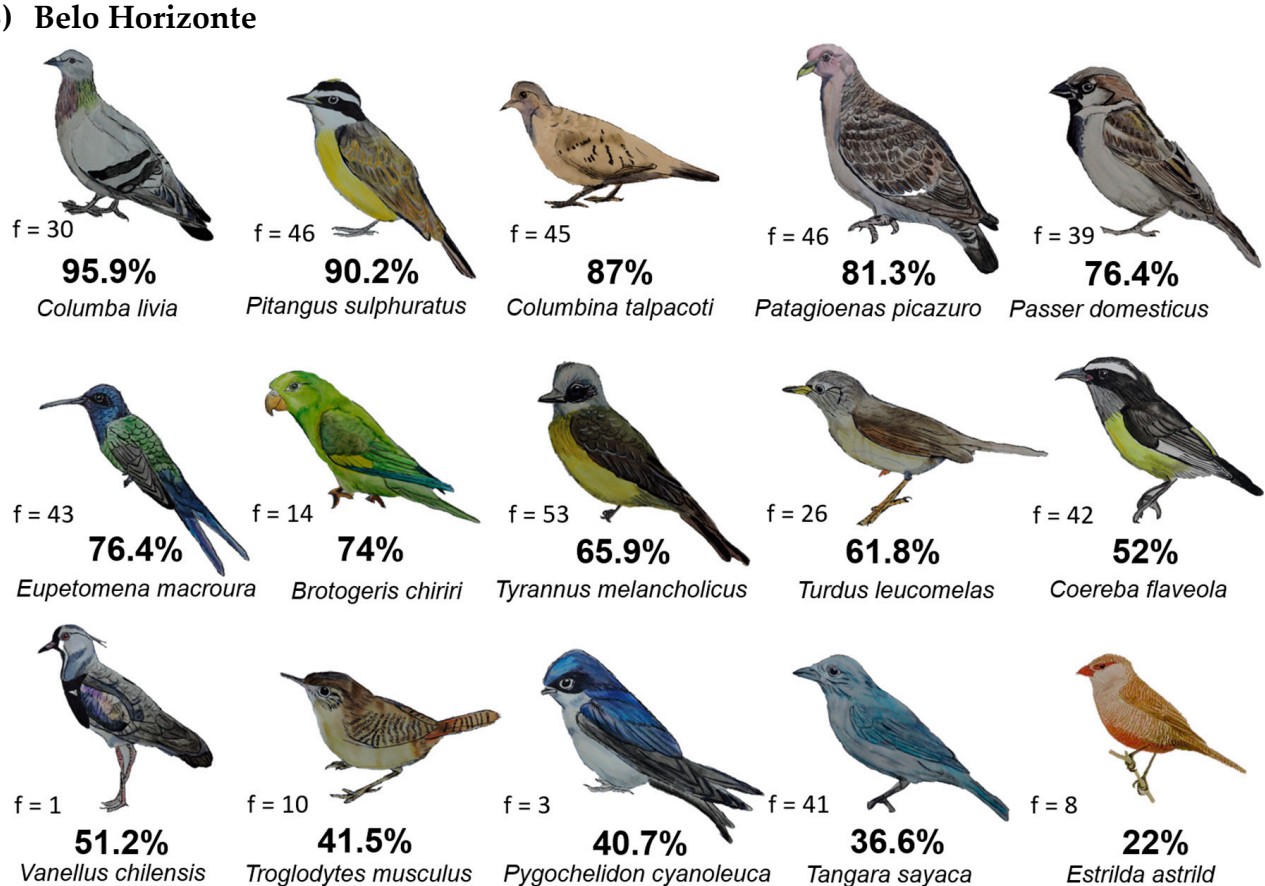

**Figure 1.** Percentage of people who observed these 15 bird species most frequently recorded during surveys in Brazilian cities: (**A**) Bauru (São Paulo) and (**B**) Belo Horizonte (Minas Gerais). The percentage represents birds that are the most seen by people daily and f means the frequency of each species recorded during our surveys. The bird's pictures are watercolors painted by Gabriela Rosa based on scientific illustrations from the *Handbook of the Birds of the World* (HBW Alive).

In Belo Horizonte, the birds most daily seen by people were the Domestic Pigeon (*C. livia*), Great Kiskadee (*P. sulphuratus*), Ruddy Ground Dove (*Columbina talpacoti*), Picazuro Pigeon (*Patagioenas picazuro*) and House Sparrow (*P. domesticus*). These species are exactly the most frequent bird species recorded during the surveys conducted in Belo Horizonte [30]: *C. talpacoti*, 594 records; *C. livia*, 490; *P. picazuro*, 386; *P. domesticus*, 350; *P. sulphuratus*, 345; Figure 1B).

The results also confirmed that, in both cities, the least frequent bird species recorded during bird surveys of each city according to our bird data were also the ones that people saw the least (Figure 2).

Considering bird songs, most respondents had already heard at least some of the most frequent ones among those presented in the questionnaire (Bauru: 88.4%; Belo Horizonte: 91.9%). Only a few people declared that they had not heard any song (Bauru: 11.6% and Belo Horizonte: 8.1%). In both cities, the Great Kiskadee (*Pitangus sulphuratus*) was the most heard and one of the most seen bird species. The Picazuro Pigeon (*Patagioenas picazuro*) and House Sparrow (*Passer domesticus*) appear in second and third place of the most heard bird species, depending on the city (Figure 3).

## A) Bauru

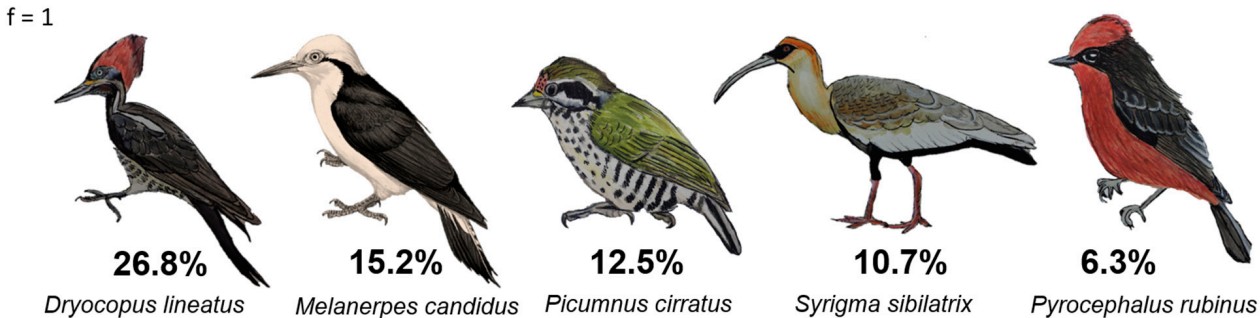

f = 1

| 26.8% | 15.2% | 12.5% | 10.7% | 6.3% |
| *Dryocopus lineatus* | *Melanerpes candidus* | *Picumnus cirratus* | *Syrigma sibilatrix* | *Pyrocephalus rubinus* |

## B) Belo Horizonte

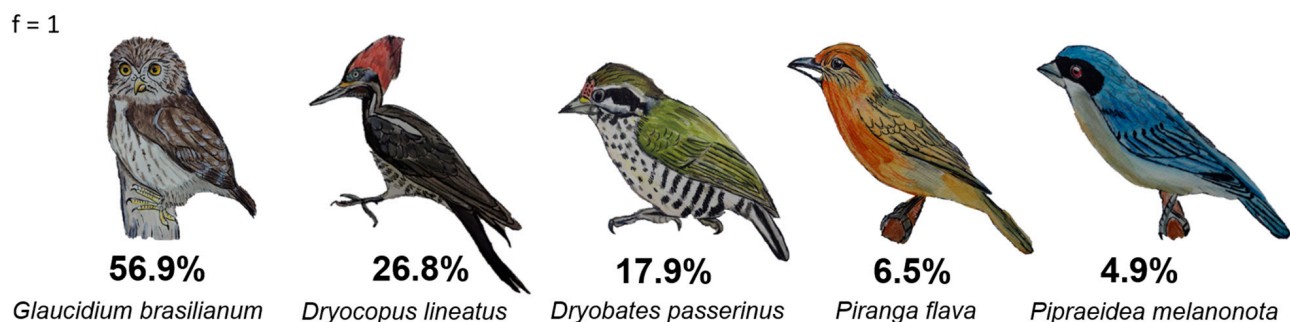

f = 1

| 56.9% | 26.8% | 17.9% | 6.5% | 4.9% |
| *Glaucidium brasilianum* | *Dryocopus lineatus* | *Dryobates passerinus* | *Piranga flava* | *Pipraeidea melanonota* |

**Figure 2.** Percentage of people who had seen the 5 bird species least frequently recorded during surveys in Brazilian cities: (**A**) Bauru (São Paulo) and (**B**) Belo Horizonte (Minas Gerais). The percentage represents birds that are the most seen by people daily and f means the frequency of each species recorded during our surveys. The bird's pictures are watercolors painted by Gabriela Rosa based on scientific illustrations from the *Handbook of the Birds of the World* (HBW Alive).

Regarding birds that live in urban areas, almost half of respondents (47.3%) in Bauru and 30.9% in Belo Horizonte believed that only fewer than 25% of species can live in each urban area. In reality, this percentage is higher than perceived by the respondents: in Bauru, we recorded 36% of the species that occur in the municipality (which includes natural and agricultural areas) in our bird survey (according to the Wikiaves records database); and in Belo Horizonte, 41.24% of birds observed in the municipality territory appeared in the streets. This result showed that people have a low awareness of the number of bird species that may live near them. In fact, only 16.1% of respondents in Bauru and 19.5% in Belo Horizonte noted the correct percentages.

Finally, the word cloud analysis showed the main feelings that respondents associated with urban birds. In Bauru, the most frequent words were admiration (n = 7), happiness (n = 7), beauty (n = 5), life (n = 5), wonderful (n = 4), peace (n = 3), freedom (n = 3), tranquility (n = 3) and worry (n = 3), where n represents the number of people who answered it (Figure 4A). In Belo Horizonte, the most frequent words were admiration (n = 13), happiness (n = 10), beauty (n = 8), love (n = 4), important (n = 4), and peace (n = 3) (Figure 4B). The interesting thing to note is that the three most cited words were the same in both cities (Figure 4). Another point that we noticed is that the feelings towards birds were mostly positive, and only a few people also mentioned feelings like disgust, worry, and illness.

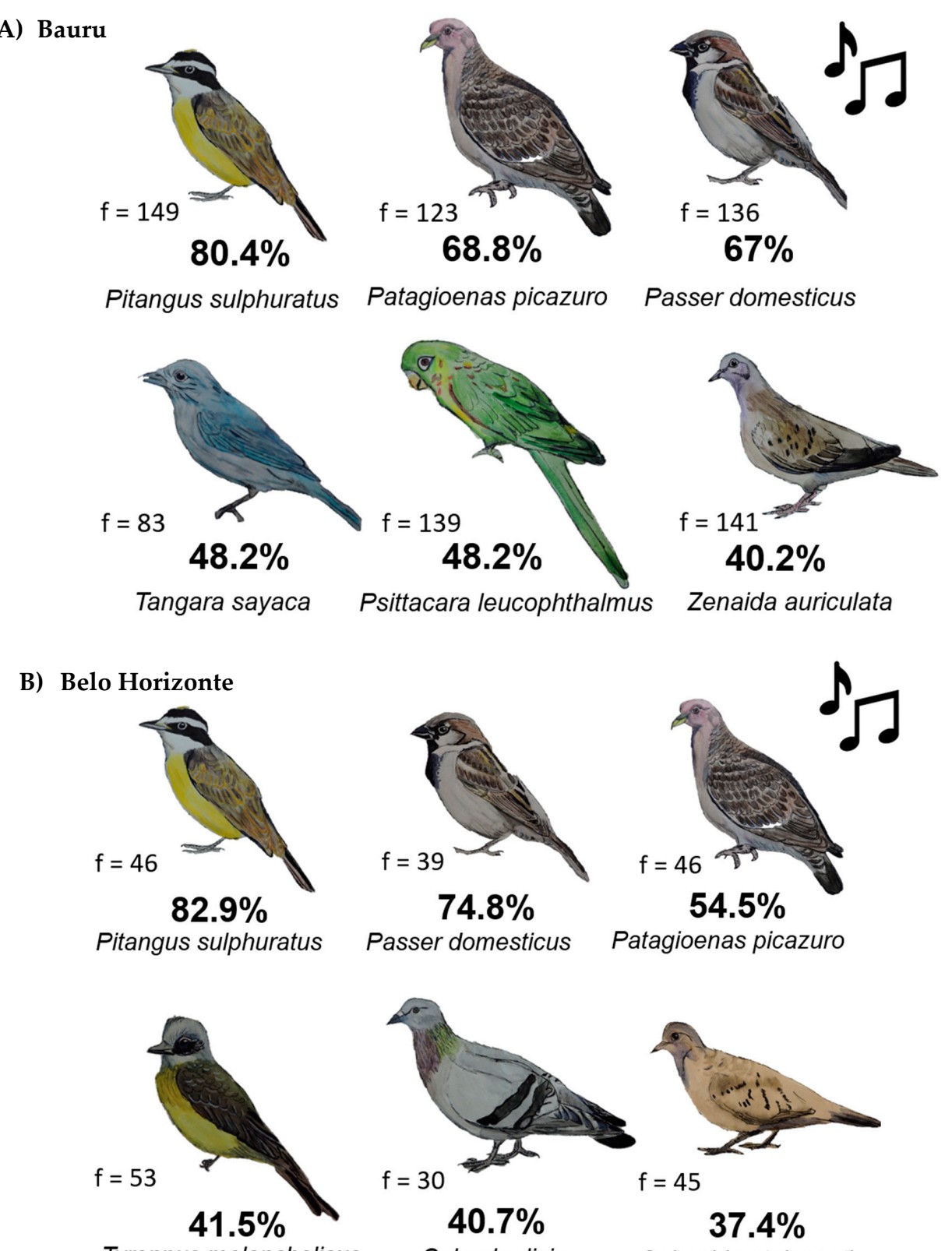

**Figure 3.** Percentage of people who have heard the song of these bird species in Brazilian cities: (**A**) Bauru (São Paulo) and (**B**) Belo Horizonte (Minas Gerais). The percentage represents birds that are the most seen by people daily and f means the frequency of each species recorded during our surveys. The bird's pictures are watercolors painted by Gabriela Rosa based on scientific illustrations from the *Handbook of the Birds of the World* (HBW Alive).

**A) Bauru**

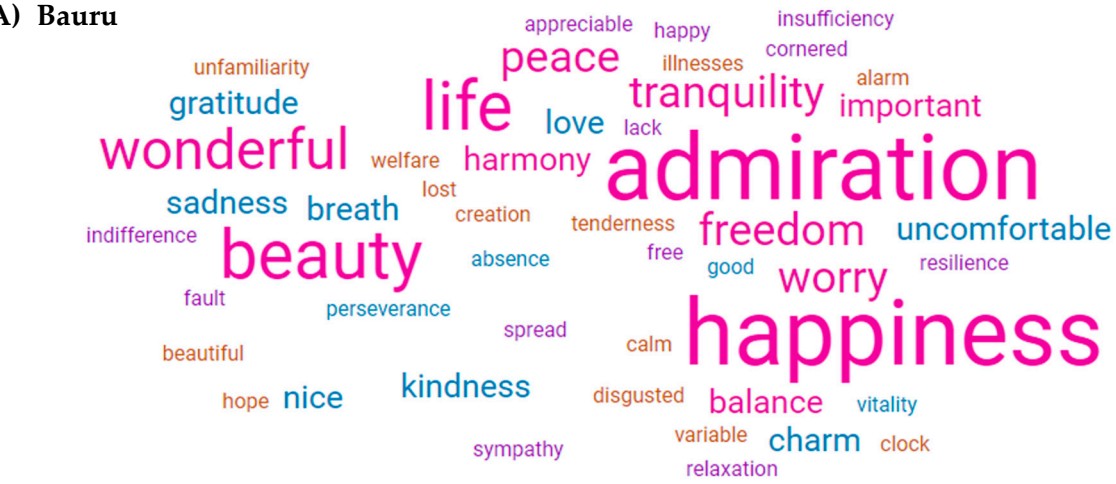

**B) Belo Horizonte**

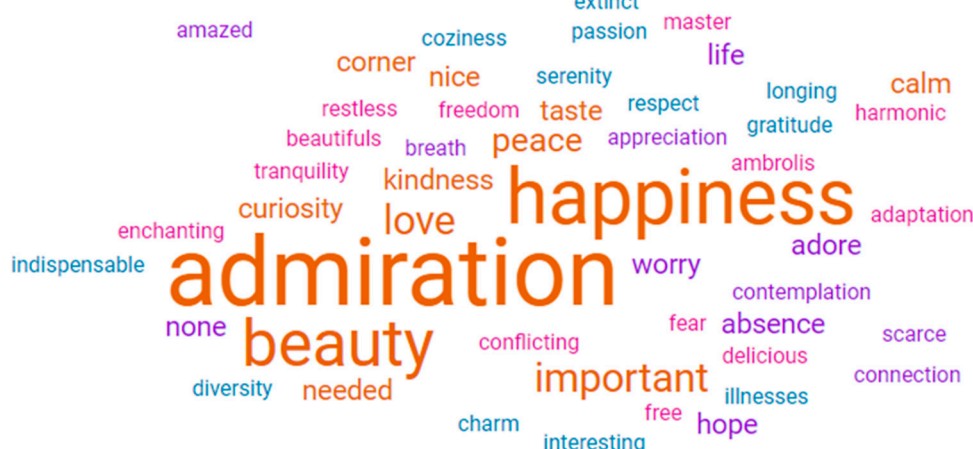

**Figure 4.** Word cloud analysis of the main feelings that respondents associate with urban birds in two Bra-zilian cities: (**A**) Bauru (São Paulo) and (**B**) Belo Horizonte (Minas Gerais).

## 4. Discussion

Our study confirmed that most respondents were aware of the importance of birds to ecological balance and have a generally positive attitude towards most of the bird species. Also, our study provides evidence that people with higher education backgrounds perceive and can recognize some of the most common bird species in each city.

### 4.1. Social Context

Unfortunately, the responses collected did not proportionately represent the population of these cities because these questionnaires only reached the group of people who were willing to respond online. Because the present research was carried out during the COVID-19 pandemic, it was difficult to access different sociodemographic groups. Since we largely depended on social media, our approach ended up selecting a specific group.

In previous studies conducted by our research group [40–43], people were interviewed based on the age and social proportion of the population. In other words, if the population had 30% elderly individuals, 30% of the respondents had to be elderly. If there were 56% women in the population, 56% of the interviewees were women. If 40% had a low income, 40% of the respondents had to have a low income. However, to achieve these proportions, we needed to perform in-person interviews with specific groups of people who were not accessible through social media. Unfortunately, elderly individuals and those with a low income usually do not respond to online surveys, maybe due to unfamiliarity with the

importance of the research or because it is something very distant from their reality. The online survey method yielded responses that did not accurately represent the population of these cities, but it did provide interesting insights into how the population with higher education background perceives urban birds.

### 4.2. People Perception of the Importance of Birds

Our research substantiated that most participants recognize the significance of birds in maintaining ecological balance. When people were asked which benefits they attribute to birds in cities, most respondents mentioned seed dispersal and biological control in both cities. Birds play vital roles as seed dispersers in human-altered landscapes, helping to maintain and restore plant communities [22,24,25]. Flower pollination and ecological balance were also two benefits frequently mentioned by people, confirming that most people with higher education backgrounds perceive and understand the importance of birds.

Overall, our study provides evidence that people with higher education backgrounds perceive and can recognize some of the most common bird species in each city. Most of the species presented in our questionnaire are common in urban environments in southeast Brazil and respondents were familiar with a high number of them. In Bauru, four out of five birds most seen by respondents were also the ones most frequently observed through our bird survey. In Belo Horizonte, the birds most seen by respondents were exactly the most frequent according to the literature [30]. This is an interesting result because the research was conducted mainly across streets (not in green areas or urban parks), thus people probably have more contact with these species daily, which may explain the greater correlation between the results in Belo Horizonte.

However, we observed that respondents underestimated the number of birds that can live in these urban areas. This happens because human–bird encounters are influenced by the visibility duration and obstruction of each species. A recent study found that bird species with lower visibility should be less well-known by the general public and may have a lower impact on recreational values [44]. Also, the song of birds is still a sense less experienced and perceived by people. This this could probably happen because of the noise pollution (mainly traffic noise) and the highly dynamic urban life that makes it difficult for some people to hear or notice bird songs within cities. Despite most respondents being able to recognize them, when compared to vision, people experience the hearing sense less. This result may indicate the need to expand environmental education initiatives. Moreover, the implementation of multifunctional ecological corridors [45] and other initiatives such as birdwatching events [46] may assist in increasing people's awareness about the bird species that live around them, conciliating the coexistence between people and biodiversity in cities.

### 4.3. How People Feel about Birds

We also found that respondents had a generally positive attitude towards most of the bird species. Considering social aspects, most birds within cities provide human connection with nature, life, joy, beauty, and well-being. Many people in Bauru and Belo Horizonte mentioned these as the main benefits provided by birds. Research in the field of environmental psychology has shown that exposure to natural systems positively affects human well-being and health [9,20,47,48]. Other studies conducted in Australia and South Africa also found that most people have positive attitudes towards birds [11,21,49]. However, there is a knowledge gap in research examining how people perceive urban wildlife, and the range of animals studied remains severely restricted [10,14,15]. Therefore, it would be interesting to carry out more studies focusing on other audiences and social groups in other cities, regions, and countries to assess if people who live under a variety of urban conditions have similar perceptions about birds.

However, there is a clear difference in human attitude and perception according to species. While the majority of people associated most bird species with positive words—such

as admiration, happiness, and beauty—they also disliked exotic species such as the domestic pigeon and the house sparrow. These species are associated with disease, dirt, disgust, and ecological imbalance, according to respondents. This result is similar to other research, in which it was observed that particular species are more appreciated (e.g., squirrels) than others (e.g., arthropods) [10,21].

*4.4. The Importance of Studying Human Perception of Animals*

Understanding how humans perceive animals is important for creating a conservation agenda and planning urban landscapes that allow a successful human-wildlife co-existence [10,11,50]. Through comprehending the societal perceptions of various animal groups, we would have information to formulate focused educational strategies aimed at bridging knowledge gaps regarding the significance of each species in preserving the equilibrium of ecosystems. Also, these studies can provide insights into potential environmental events and lectures focused on bird species that are related to a negative perception among the public, such as pigeons and sparrows. In this way, it is essential to involve a mix of researchers, practitioners, policymakers, and urban planners together with citizen support to create strategies for the better management of urban wildlife [2,51].

Nevertheless, it is important to mention that the scope of this study was limited to people who had at least an undergraduate degree. The present research was carried out during the pandemic, which reduced the access to different sociodemographic groups since we were unable to conduct in-person interviews. Despite the questionnaires being sent to various email addresses and posted in social media groups that encompass a significant portion of the population from different social groups, most of the individuals who volunteered to respond were individuals with a high level of education. Ideally, the number of questionnaires should have a sample size that represents the population and should be applied in person to reach people of different social classes, ages, and genders, following the same proportionality as population data of each municipality, as performed in many studies [40–43]. Furthermore, the choice of interviewees must be random but based on the age and sex proportion of the original population [52]. Despite this, our study provides interesting evidence about the human perception of birds, an interesting field of study that deserves to be deepened.

Birds are part of the urban landscape and stimulate human senses. They can bring good sensations and feelings and can increase the connection of humans with nature in urban environments. A highly 'imaginable' city would invite our eyes and ears to engage in an active participation [53]. The city exists through bodily experience, which is multisensory [54]. Thus, studying birds and perceiving them as part of the urban landscape can be useful to stimulate the imaginability of cities and, at the same time, contribute to the conservation of different groups of animals that are important for the ecological balance of urban ecosystems [55].

**5. Conclusions**

We showed that people with undergraduate backgrounds can recognize the most frequent bird species and are aware of the ecological importance of birds for the balance of urban ecosystems. However, most people underestimate how many bird species can live in urban areas. Also, we saw that most people have already heard some bird songs, but this sense is less experienced compared to the visual one. Regarding the feelings and attitudes toward birds, we showed that most people associate most bird species with positive sensations such as beauty, joy, well-being, and peace. On the other hand, species such as the domestic dove are considered pests and generate negative sensations.

The concept of environmental perception seeks to cover aspects that influence the natural, physical, and humanized environment through attitudes, values, and worldviews. By understanding how society perceives different animal groups, we can develop targeted educational strategies to fill knowledge gaps about the importance of each species. Finally, the positive perceptions of birds among respondents raise the question of whether such

perceptions could be utilized in assessing the effectiveness of citizen science and conservation efforts. Exploring this aspect, either local or in other contexts, could enrich future research directions about how people perceive animals in cities.

**Supplementary Materials:** The following supporting information can be downloaded at: https://www.mdpi.com/article/10.3390/birds5020014/s1, Figure S1: Percentage of strongly agreement with the questionaries' statements; Figure S2: Damages that may be caused by birds according to respondents from Belo Horizonte (Minas Gerais, Brazil) and Bauru (São Paulo, Brazil); Figure S3: Benefits associated with birds in cities mentioned by respondents from Belo Horizonte (Minas Gerais, Brazil) and Bauru (São Paulo, Brasil). The number means how many times each benefit was mentioned and not how much people said it, once many people mentioned more than one benefit.

**Author Contributions:** Conceptualization, G.R.G., M.C.R. and J.C.P.; data curation, G.R.G.; formal analysis, G.R.G. and M.C.R.; funding acquisition, G.R.G., M.C.R. and J.C.P.; investigation, G.R.G., M.C.R. and J.C.P.; methodology, G.R.G., M.C.R. and J.C.P.; project administration, G.R.G.; resources, G.R.G., M.C.R. and J.C.P.; software, G.R.G.; supervision, J.C.P.; validation, G.R.G.; visualization, G.R.G., M.C.R. and J.C.P.; writing—original draft, G.R.G.; writing—review and editing, G.R.G., M.C.R. and J.C.P. All authors have read and agreed to the published version of the manuscript.

**Funding:** This research was funded by Coordenação de Aperfeiçoamento de Pessoal de Nível Superior—Brasil (CAPES), the São Paulo Research Foundation (grants #2013/50421-2; 2020/01779-5; 2021/08322-3; 2021/08534-0; 2021/10195-0; 2021/10639-5; 2022/10760-1; 2018/00107-3), and the Conselho Nacional de Desenvolvimento Científico e Tecnológico (grants #442147/2020-1; 440145/2022-8; 402765/2021-4; 313016/2021-6; 440145/2022-8).

**Institutional Review Board Statement:** Not applicable.

**Data Availability Statement:** Data is contained within the article or Supplementary Material.

**Acknowledgments:** M.C.R. thanks to the Sao Paulo Research Foundation—FAPESP (processes #2013/50421-2; #2020/01779-5; #2021/06668-0; #2021/08322-3; #2021/08534-0; #2021/10195-0; #2021/10639-5; #2022/10760-1) and National Council for Scientific and Technological Development—CNPq (processes #442147/2020-1; #402765/2021-4; #313016/2021-6; #440145/2022-8; 420094/2023-7), and Sao Paulo State University—UNESP for their financial support. This study is also part of the Center for Research on Biodiversity Dynamics and Climate Change, which is financed by the Sao Paulo Research Foundation—FAPESP.

**Conflicts of Interest:** The authors declare no conflict of interest.

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
