# Peer review of "Human Perception of Birds in Two Brazilian Cities"

_2673-6004, doi:10.3390/birds5020014_

Round 1
Reviewer 1 Report
Comments and Suggestions for Authors
O trabalho dá uma rica contribuição para a importância do conhecimento dos cidadãos sobre a biodiversidade das áreas urbanas. Demonstra que o reconhecimento da convivência faz do cidadão um agente de conservação. Metodologia clara e resultados relevantes
Author Response
Dear Reviewer,
Thank you for your comments and feedback. We have made improvements to our manuscript, and all changes have been highlighted in red throughout the text, which is attached to this message.
Best regards

Reviewer 2 Report
Comments and Suggestions for Authors
Thank you for this effort. I have made many comments and suggestions in the attached pdf document of the manuscript.

The English would improve with an editor who is a native speaker. Overall it is understandable but in several instances inappropriate terms were used, as noted.
Author Response
Dear Reviewer,
Thank you for your comments and feedback. We have made improvements to our manuscript, and all changes have been highlighted in red throughout the text. Our responses to the comments are provided in the attached PDF
Best regards

Reviewer 3 Report
Comments and Suggestions for Authors
In this study, entitled “Human perception of birds in two Brazilian cities”, the authors conducted a survey on people’s perception on local birds in two urban areas in Brazil. While the data may be worth to be published, the study has several limitations that I’d like to outline here.
I’m confused about the rationale behind focusing on perceptions (lines 50-70). Apparently, the point is to emphasize that these are linked to feelings and thus are irrational. I’m not very afraid to state however that all perceptions are mediated by the unconscious and thus are at least in part influenced by our value systems. This is perfectly exemplified by the divergent results reported for native and non-native species’ perceptions, which clearly have a social basis. I’d suggest to reducing this section which is not fundamental to understanding the study’s results.
I also think that site selection needs to be properly justified. Why these two cities and why results must differ between them? These two areas are introduced but it would be necessary to explain carefully why these two cities are particularly relevant or why it is interesting comparing results from these two areas.
It is said and emphasized that people’s perceptions on native and exotic species differ, but the quantitative data is not shown. I actually think that it would be important to perform separate analyses of this issue, to clearly show the readers that there was a significant difference in how these two groups of species were perceived by people.
Regarding species ID by people, species conspicuousness (size, song, color, behavior, habitat use) must have influenced results. This issue is completely neglected in the present study. I’d suggest incorporating these characteristics in the analysis of people’s perceptions in some way, ideally in a quantitative way, but at least the authors need to acknowledge this issue.
Results reporting can be optimized. Right now this section uses a lot of species. While I understand the interest in showing the beautiful drawings of the studied bird species, I think that overall this section is repetitive and rather uninteresting unless the authors are able to summarize key results in some way. For instance, detailed results could be provided in tables, which would help the readers quickly grasp the main results, and a few figures could be used to emphasize the most important results according to the authors’ point of view.
In figures 4-9, what is “f”?
Author Response
Dear Reviewer,
Thank you for your comments and feedback. We have made improvements to our manuscript, and all changes have been highlighted in red throughout the text, which is attached to this message.
Below are the comments for each point mentioned:
"I’m confused about the rationale behind focusing on perceptions (lines 50-70). Apparently, the point is to emphasize that these are linked to feelings and thus are irrational. I’m not very afraid to state however that all perceptions are mediated by the unconscious and thus are at least in part influenced by our value systems. This is perfectly exemplified by the divergent results reported for native and non-native species’ perceptions, which clearly have a social basis. I’d suggest to reducing this section which is not fundamental to understanding the study’s results."
We reduced this section and improved this paragraph
"I also think that site selection needs to be properly justified. Why these two cities and why results must differ between them? These two areas are introduced but it would be necessary to explain carefully why these two cities are particularly relevant or why it is interesting comparing results from these two areas."
We added the explanation about why these two cities and also improved the method description.
"It is said and emphasized that people’s perceptions on native and exotic species differ, but the quantitative data is not shown. I actually think that it would be important to perform separate analyses of this issue, to clearly show the readers that there was a significant difference in how these two groups of species were perceived by people."
These data were derived from open-ended responses, meaning there was no separate analysis. As there are only two exotic bird species, a quantitative analysis with a suitable sample size was not feasible. Furthermore, this result was unexpected; it emerged when some individuals mentioned that positive sentiments did not apply to pigeons and sparrows. We counted how many people mentioned it and conducted this analysis based on it.
"Regarding species ID by people, species conspicuousness (size, song, color, behavior, habitat use) must have influenced results. This issue is completely neglected in the present study. I’d suggest incorporating these characteristics in the analysis of people’s perceptions in some way, ideally in a quantitative way, but at least the authors need to acknowledge this issue."
We compared the results of the questionnaires with the data collected across each city. The most frequent bird species observed by the respondents are practically the same ones most frequently observed in both cities. Thus, we believe that the bias regarding the size of the bird species is not influencing these results, since both bird surveys were conducted by experienced ornithologists.
Results reporting can be optimized. Right now this section uses a lot of species. While I understand the interest in showing the beautiful drawings of the studied bird species, I think that overall this section is repetitive and rather uninteresting unless the authors are able to summarize key results in some way. For instance, detailed results could be provided in tables, which would help the readers quickly grasp the main results, and a few figures could be used to emphasize the most important results according to the authors’ point of view.
We deleted 3 figures and we consolidated six figures into just three. Also we improved this section.
"In figures 4-9, what is “f”?
We explained what the term "frequency" means in section 2.3. Nevertheless, we refined the explanation and improved these sentences.
Best regards,

Round 2
Reviewer 2 Report
Comments and Suggestions for Authors
The authors have answered my questions and addressed my concerns as provided in my first review. This is a great improvement, thank you. I just have a few additional recommendations:
1 - There are some language issues throughout the paper that require attention from a professional English editor; otherwise some of the language is confusing and some of the findings will be lost to readers due to word choice. I have made some specific notes on English issues in a few areas throughout the paper in my attached draft with feedback but the entire paper needs to be edited by a professional editor; I have also made a few other suggestions in the attached draft, including the below.
2 - I believe the "results" section should include an additional paragraph or two, ideally just before the "conclusion" section putting the results in global context. The authors do a fine job discussing the results in the context of their study, but how do they compare with other studies of people's perceptions of birds in other regions of the world, such as elsewhere in South America, in North America, Europe, Asia, etc? Adding and discussing some relevant studies from elsewhere to compare with this study would deepen and strengthen the authors' presentation of findings from this study.
3 - Encouraged by the positive perceptions of birds by many respondents in this study, I wondered whether this factor might be world considering in measuring the success of citizen science and conservation initiatives. Has this been done, locally or elsewhere, or might it be done? If so, it would be good to include a sentence or two discussing this and if not, this might be a suggestion you make for future study. In either case, I think the discussion/conclusions section would be improved by adding a sentence or two with the next questions you believe are worthy of study to build on the findings presented in this paper, to guide future research and/or conservation initiatives.

There are some language issues throughout the paper that require attention from a professional English editor who understands the study (i.e., ideally an editor who is native English speaker and scientist); otherwise some of the language is confusing and some of the findings will be lost to readers due to word choice. Some specific notes and suggestions to this effect are included in the attached draft.
Author Response
1 - There are some language issues throughout the paper that require attention from a professional English editor; otherwise some of the language is confusing and some of the findings will be lost to readers due to word choice. I have made some specific notes on English issues in a few areas throughout the paper in my attached draft with feedback but the entire paper needs to be edited by a professional editor; I have also made a few other suggestions in the attached draft, including the below.
We have corrected and accepted all suggestions. We will also perform a professional English review.
2 - I believe the "results" section should include an additional paragraph or two, ideally just before the "conclusion" section putting the results in global context. The authors do a fine job discussing the results in the context of their study, but how do they compare with other studies of people's perceptions of birds in other regions of the world, such as elsewhere in South America, in North America, Europe, Asia, etc? Adding and discussing some relevant studies from elsewhere to compare with this study would deepen and strengthen the authors' presentation of findings from this study.
We added sentences about that.
3 - Encouraged by the positive perceptions of birds by many respondents in this study, I wondered whether this factor might be world considering in measuring the success of citizen science and conservation initiatives. Has this been done, locally or elsewhere, or might it be done? If so, it would be good to include a sentence or two discussing this and if not, this might be a suggestion you make for future study. In either case, I think the discussion/conclusions section would be improved by adding a sentence or two with the next questions you believe are worthy of study to build on the finding
We added sentences about that.

Reviewer 3 Report
Comments and Suggestions for Authors
I have no further comments or suggestions.
Author Response
Thank you for your review. We hope we are now submitting an improved version of the manuscript.